# Hierarchical Structuring of Black Silicon Wafers by Ion-Flow-Stimulated Roughening Transition: Fundamentals and Applications for Photovoltaics

**DOI:** 10.3390/nano13192715

**Published:** 2023-10-06

**Authors:** Vyacheslav N. Gorshkov, Mykola O. Stretovych, Valerii F. Semeniuk, Mikhail P. Kruglenko, Nadiia I. Semeniuk, Victor I. Styopkin, Alexander M. Gabovich, Gernot K. Boiger

**Affiliations:** 1Igor Sikorsky Kyiv Polytechnic Institute, National Technical University of Ukraine, Prospect Beresteiskyi, 37, 03056 Kyiv, Ukraine; stretovychmykola@gmail.com; 2G.V. Kurdyumov Institute for Metal Physics, National Academy of Sciences of Ukraine, 36 Academician Vernadsky Boulevard, 03142 Kyiv, Ukraine; 3Department of Mechanical and Aerospace Engineering, University of Liverpool, Liverpool L69 3GH, UK; 4Institute of Physics of the Ukrainian National Academy of Sciences, Nauka Avenue, 46, 03028 Kyiv, Ukraine; valeriisemeniuk7@gmail.com (V.F.S.); mkruglenko@ukr.net (M.P.K.); vstyopkin@gmail.com (V.I.S.); alexander.gabovich@gmail.com (A.M.G.); 5GreSem Innovation LLC, Vyzvolyteliv Avenue, 13, 02660 Kyiv, Ukraine; snadiyaukrnet@ukr.net; 6ICP Institute of Computational Physics, ZHAW Zürich University of Applied Sciences, Wildbachstrasse 21, CH-8401 Winterthur, Switzerland

**Keywords:** surface mass transfer, Monte Carlo method, helicon discharge, ion plasma flow, silicon wafer processing, black silicon

## Abstract

Ion-flow-stimulated roughening transition is a phenomenon that may prove useful in the hierarchical structuring of nanostructures. In this work, we have investigated theoretically and experimentally the surface texturing of single-crystal and multi-crystalline silicon wafers irradiated using ion-beam flows. In contrast to previous studies, ions had relatively low energies, whereas flow densities were high enough to induce a quasi-liquid state in the upper silicon layers. The resulting surface modifications reduced the wafer light reflectance to values characteristic of black silicon, widely used in solar energetics. Features of nanostructures on different faces of silicon single crystals were studied numerically based on the mesoscopic Monte Carlo model. We established that the formation of nano-pyramids, ridges, and twisting dune-like structures is due to the stimulated roughening transition effect. The aforementioned variety of modified surface morphologies arises due to the fact that the effects of stimulated surface diffusion of atoms and re-deposition of free atoms on the wafer surface from the near-surface region are manifested to different degrees on different Si faces. It is these two factors that determine the selection of the allowable “trajectories” (evolution paths) of the thermodynamic system along which its Helmholtz free energy, *F*, decreases, concomitant with an increase in the surface area of the wafer and the corresponding changes in its internal energy, U (dU>0), and entropy, S (dS>0), so that dF=dU – TdS<0, where T is the absolute temperature. The basic theoretical concepts developed were confirmed in experimental studies, the results of which showed that our method could produce, abundantly, black silicon wafers in an environmentally friendly manner compared to traditional chemical etching.

## 1. Introduction

Silicon is the most favored material in producing solar cells of high efficiency. In the technology for direct solar energy conversion into electricity, the cell efficiency is determined by the absorption coefficients of the solar radiation [1,2,3]. The efforts of scientists and engineers are focused on maximizing these coefficients. In other words, the creation of a “black” silicon surface structure is a necessary condition to obtain highly effective solar cells [4,5,6]. Such structures should include components with typical sizes of ultraviolet, visible, and infrared light wavelengths. Furthermore, in surface processing, additional nanostructures are inevitably formed against the microstructure background [7,8,9], which form hierarchical patterns. These nanostructures also influence the optical and electrical properties of the silicon surface [10].

It is clear that these hierarchical patterns [11,12] should appear on any disordered surface of a solid material, which is affected by external stimulating factors, such as laser, electron, or ion beams [13,14,15,16,17,18,19,20,21,22,23]. The nanostructure component is crucially important since it determines the basic properties at the level of surface chemical bonds. Hence, we are especially interested in the elementary acts creating the nanostructures concerned.

The controlled fabrication of ordered nanostructures on the surface of various types of materials including metals, semiconductors, dielectrics, and polymers has been of great interest in material science for decades and has proven to give rise to or drastically improve the practical functions of different devices made of such materials [24,25,26,27,28,29,30,31,32,33]. The micro- and nano-scale objects of one-, two-, and three-dimensional characters have found use in multiple areas of scientific study and for industrial application [19,34,35,36,37,38,39,40,41,42,43].

The physical factors that determine the creation and further development of ordered structures are very diverse [44], but they all manifest themselves at high enough temperatures of the perturbed material. One may divide the corresponding temperature and impact regimes into three groups, which differ in the aggregation state of precisely those upper layers of the material, in which nano- and micro-structures emerge on an initially flat surface.

The first (hard) mode corresponds to the cases when an external action on the surface of a solid (by releasing kinetic energy of impinging particles and potential energy of plasma ions, or via laser irradiation heating) completely destroys the crystal lattice and brings its upper layers to a genuine liquid state. A striking example of such a mode realization was proposed by researchers [22]. Namely, the interaction of laser pulse series induced the development of thermo-convective instabilities, convection rolls, and capillary waves (see also [45,46,47,48]) in a molten nickel layer, which led to various modifications of the initially randomly rough surface. Depending on the number of laser pulses, the time interval between them, and their power, different forms such as self-ordered peaks, labyrinthine structures, hexagonal arrays, or oriented stripes were formed [22]. In this hard regime, local melting can occur with nano-bumps created and nano-jets emitted [49].

On the other hand, the third (softest) mode corresponds to surface instability in the pre-melting situation, when surface chemical bonds are locally weakened but a true melting transition is not achieved. Pre-melting is characterized by changes in crystal physical properties far below the melting point [50]. There is an alternative viewpoint that pre-melting means true melting of pure crystals in certain small regions and over small durations [51,52]. Of course, such a scenario is also possible but in the hard mode characterized above rather than in the soft regime. The latter was studied in detail when spontaneous disintegration of nanowires was observed while maintaining their crystal lattice. Platinum [53], copper [54,55], gold [56], and silver [57] nanowires can break up into a chain of nanodrops at temperatures of approximately 600 °C, 400 °C, 300 °C, and ~20 °C, respectively, noticeably below the melting points of metals from which they were synthesized. The developing thermal instability is an apparent analogue of the Rayleigh instability of liquid cylindrical jets [58,59,60]. However, in contrast to a liquid with its emerging internal flows, periodic cross-section modulations of the initially cylindrical nanowire are associated with the surface diffusion of atoms from zones with a low binding energy (high surface curvature) to zones with a higher binding energy (low surface curvature).

In the classical model of surface diffusion [61], suggested under the assumption that the surface energy density is isotropic, the development of nanowire constrictions is accompanied by a decrease in the lateral surface area (and the concomitant decrease in surface energy, $U_{surf}$). Similar to the results of the Rayleigh process [58,59], only periodic perturbations of the nanowire radius with a wavelength $\lambda >\lambda_{cr}=2\pi r_{0}$ grow in time, with the maximum increment corresponding to the wavelengths $\lambda_{R}\approx 9r_{0}$, where $r_{0}$ is the initial radius of a nanowire or liquid jet. Moreover, if one takes into account the anisotropy of the surface energy density [62,63], the observed breakup of gold nanowires into fragments of length, *l* within the range of $7.5≲l/r_{0}≲$ 30 can be explained [57,64,65,66]. However, the breakup of Si nanowires into fragments of lengths below the instability threshold of $l\approx 4.5r_{0}<\lambda_{cr}$ [67] should be associated, as was shown in [68], with a specific manifestation of the so-called roughening transition (RT) [69,70,71,72,73,74,75].

If the temperature of a flat surface, *T*, exceeds the critical one, $T_{cr}$, then periodic perturbations (“mounds and cavities”) can develop on it, gradually increasing in height and in area. The RT effect occurs only on selected crystal faces. Its driving mechanisms are the surface diffusion of atoms and re-deposition on the surface of atoms evaporated from the upper layer (layers). The formed mounds create shadow zones around themselves for atoms settling in the diffusion mode and concentrate their influxes on mound tops. This positive feedback contributes to the further development of the initial perturbations on the surface. With such a surface modification, the surface energy of the system, *U_surf_*, increases. However, the developing roughening at a finite temperature does not contradict the first law of thermodynamics, since the free energy, *F*, of the system, which determines the direction of kinetic processes, decreases: $dF=dU_{surf}-TdS<0$. Here, *S* is the entropy of the system (dS>0 when the surface area increases).

Roughening in the evolution of the Si nanowire morphology manifests itself only when the wire diameters are sufficiently large,$\;d_{0}=2r_{0}\geq 8\;\mathrm{nm}$, i.e., in the initial stages when the nanowire lateral surface consists of rather wide bands (faces), on which the RT can develop. Short-wavelength, ($\lambda /r_{0}\sim 4.2$), long-lived quasi-stable modulations of the Si-nanowire [67,68] are induced only when its axis is oriented in the [111] direction with respect to the internal crystal structure. For the [100] orientation, the nanowire splits into fragments with the approximate length of $l\approx 18r_{0}$ [68].

Note that one can also induce RT by external irradiation of the surface. If at elevated temperatures surface perturbations do not develop on a given face, then stimulation of the surface diffusion by such irradiation can lead to the excitation of the desired instability. The results of [76] constitute a striking example of this phenomenon. Specifically, when irradiated with electrons, a gold nanowire with the [110] orientation, which is the most resistant to fragmentation, formed pronounced stable modulations of its radius with a wavelength of $\lambda \sim 5.6r_{0}<\lambda_{cr}$. However, the results of experiments [77] for an Ag nanowire with the [111] orientation turned out to be unexpected. The surface diffusion of its atoms was stimulated by argon ions Ar^+^ that caused the nanowire to disintegrate into fragments of length $l\sim 16r_{0}$ instead of $l\sim 8r_{0}$, which would develop in the absence of irradiation. A theoretical analysis of the various observed responses to the external irradiation of nanowires with a face-centered cubic crystal lattice was carried out by researchers in [78,79].

The second mode, which has an intermediate strength of external influence, i.e., its strength lies between the first (hard) and the third (soft) ones, assumes a moderate influence of external factors on the solid surface in order to ensure a controlled synthesis of micro- and nano-structures of the desired morphology. Independent of the origin of the perturbing force, the surfaces are in a non-equilibrium state and their evolution is highly non-linear [80]. In these unstable conditions, various kinds of patterns emerge including periodic ones.

Ion bombardment is the most powerful and the most frequently used method of surface modification, including sputtering and texturing [16,81]. For large energies (by convention, such energies, *E*, are larger than several hundred eV, i.e., they substantially exceed the bulk or surface cohesive energies per atom, being less than or equal to 10 eV [82,83,84]) of the impinging particles, cascade sputtering and concomitant heat release occur [85].

Depending on the particle energy, incidence angle, and surface crystallography, regular or irregular surface patterns are formed [86,87,88,89]. In particular, the structuring of the surface of semiconductors (InP, InSb) bombarded with Ar^+^ and Xe^+^ ions was achieved at ion energies $E_{ion}\;$in the range 500 eV to 2–5 keV and current densities up to *j_ion_* = 150 μA cm^−2^. However, the formation of ordered structures on the surface of metals (Cu, Al, Ni) can be achieved in the intermediate regime, as we showed earlier [90], at much lower ion energies, $E_{ion}$ = 20–30 eV, but at noticeably higher current densities, $j_{ion}\approx 50\;\mathrm{mA}/{\mathrm{cm}}^{2}$. For these parameters of the bombardment, the energy (heat) transfer to the target is significant but localized and restricted to its upper layers. The authors of [90] called such a type of ion-plasma surface engineering the ‘trampoline scenario’ and described its characteristic manifestations. The origin of the observed trampoline effect is the local creation of the non-equilibrium quasi-liquid.

In this article, we applied the developed ion-plasma technique to process the silicon surface in order to obtain black silicon and study in detail all possible structures to choose the optimal experimental parameters. To achieve this goal, we also made theoretical calculations of the elementary processes leading to the formation of beneficial nano- and micro-structures. Our approach is based on our ideas and results developed earlier [77,78,91]. One of the important factors responsible for the formation of self-ordered surface structures is the re-precipitation of atoms or ions that have previously evaporated from the surface. The process of self-ordering during the development of corresponding growth instabilities can be controlled from the outside [91,92,93]. For instance, it was shown that a controlled time-varying supply of free Si/Ge atoms to a Si/Ge nanowire surface can restrain the development of neighboring nanocluster inhomogeneity, ensuring size monodispersity and dense substrate filling by newly formed clusters.

Taking into account the above possibilities, in this work we propose to solve the problem of creating black silicon in two stages. At the initial stage, a relatively mild stimulation of surface diffusion by ions will lead to the synthesis of ordered nano-protrusions. A smooth transition to a more severe irradiation regime will provide more intense evaporation of atoms and, accordingly, increase the density of their reverse re-precipitation flux. This flux will concentrate on the apices of the formed protrusions, which will lead to a further increase in the height amplitude of the surface modulations. Recall that a similar gradual formation of surface inhomogeneities by a series of laser pulses was used in [22]. We studied the initial stages of the inhomogeneity development on different faces of silicon wafers based on the previously developed Monte Carlo model, which is presented in Section 2. Experimental results obtained on silicon polycrystals presented below largely confirm the theoretical concepts indicated above.

The outline of the article is as follows. In Section 2, the adopted physical model is presented. Section 3 describes the numerical calculations, their results, and interpretation. The experimental part of the paper is contained in Section 4, where our method of obtaining high-quality black silicon and the resulting optical properties are presented. The discussion is given in Section 5, whereas conclusions are in Section 6.

## 2. Numerical Model

In this paper, we have performed numerical simulations of the dynamics of surface transformation in Si structures with different orientations of the initially flat face by using the mesoscopic Monte-Carlo model. This model has been diligently developed and successfully applied in previous investigations of physical processes at the nanoscale. The results obtained were in good agreement with real experimental data on the breakup of nanowires with diamond-like, face-centered cubic (fcc), and body-centered cubic (bcc) lattice structures into isomeric nanoparticles [62,63,68,94], the formation of nano-clusters and nano-pillars in nonequilibrium surface growth [95,96], and nanoparticle sintering [97,98,99,100,101]. The basic probabilistic concepts of the Monte-Carlo model used are expounded in detail in the aforementioned publications but will also be briefly presented below.

In the adopted model, the Si substrate atoms are located at the sites of the diamond-like cubic lattice and each atom interacts only with its nearest neighbors in the lattice. We trace the atomic motion only in the upper layers of the substrate, which at the initial moment of time fill either a parallelepiped of dimensions w×l×h (w,l≫h) or a disk with radius R and thickness h (R≫h). The unit of length, $l_{0}$, in our work is the distance between neighboring Si atoms in the [110]-direction.
(1)$$l_{0}=a/\sqrt{2},$$
where a is the lattice constant for the diamond cubic crystal structure.

Silicon atoms lying outside the selected regions are supposed to be immobile but interact with the adjacent mobile atoms of the computational domain. When the substrate surface is bombarded, not only is the surface diffusion intensified, but so is the evaporation of atoms. Such evaporated atoms perform a random motion in a box with reflective walls, which is adjacent to the computational domain (box height is H≫h). The unit for the measurement of time is the characteristic interval, τ, between chaotic jumps of free atoms. The length of random jumps, λ, is the same in all directions ($\lambda =l_{0}$), and the corresponding diffusion coefficient for free atoms is $D=\lambda^{2}/\left(6\tau \right)$. Bound atoms are less mobile and the frequency, ν, of their jumps is lower than the frequency of free atom jumps: ν<1/τ. It is obvious that the probability of jumps, *P*, during time τ is less than one: *P* < 1. The value of *P* differs along the substrate surface because surface atoms have different binding energies depending on the number of nearest filled vacancies in the crystal lattice.

Each Monte Carlo (MC) time step corresponds to the following procedure. With a total initial number, *N*, of atoms in the system (including both bound and free atoms), we randomly select individual atoms from this set *N* times and determine a new position of each atom in accordance with the chosen probabilistic rules explained below. Individual atoms may change their state several times during such a procedure, but on average, each atom is selected once per one Monte-Carlo time step.

The probabilistic rules of the model are as follows. For any kind of nanostructure crystal lattice, the dynamics of the process in the used Monte Carlo model depend on two parameters: *p* and α. The first one determines the probability *P* of the bounded selected atom to jump to the nearest vacancy in the crystal lattice. This probability depends on the number of nearest neighbors, *m*_0_, at the initial position,
(2)$$P=e^{-m_{0}\Delta /kT}={\left(e^{-\Delta /kT}\right)}^{m_{0}}\equiv p^{m_{0}},$$
where $p=e^{-\Delta /kT}$, Δ is the activation free-energy barrier for the pairwise interaction, k is the Boltzmann constant, and T is the absolute temperature of the substrate. We emphasize that our model neglects a possible correlation between jumps, which may be of various nature and does not conspicuously change the results [102].

In the case of $m_{0}=0$ (the chosen atom is free), a jump is made in a random direction of length λ. When colliding with the wall of the near-surface box, the atom is reflected specularly from it. It is also possible to re-deposit this atom on the substrate into a near-surface vacancy of the crystal lattice. Each site of the lattice corresponds to a spatial region confined in a Wigner–Seitz unit-lattice cell. If a free atom hops into such a cell, it becomes attached to the substrate surface at its center.

When $m_{0}>0\;(m_{0}<m_{c}$, where $m_{c}$ is the coordination number for a given lattice, $m_{c}$= 4 for Si) and the hopping occurs, then a transition to ($m_{c}-m_{0}+1)$ the final state is possible, taking into account the possibility of remaining in the same position. Naturally, for $m_{0}=m_{c}$ (all nearest lattice sites are occupied), the atom remains immobile. It is the second parameter of the model, α, that determines the probabilities $p_{target}^{\left(i\right)}$ ($i=0,\;1,\ldots ,m_{c}-m_{0}$) of each possible hopping direction (i denotes a direction),
(3)$$p_{target}^{\left(i\right)}=C\times \exp(m_{t}^{\left(i\right)}\alpha ),\;\alpha =\left|\varepsilon \right|/kT,\;C={\left(\sum_{i=0}^{m_{c}-m_{0}}\exp\left(m_{t}^{\left(i\right)}\alpha \right)\right)}^{-1}$$
where $m_{t}^{\left(i\right)}$ is the number of nearest neighbors in the following new i-th state, and ε < 0 implies the binding free energy at target sites.

The model parameters p and α change with *T* and correlate with each other according to the following relationship [100]:(4)$$p={\left(p_{0}\right)}^{\alpha /\alpha_{0}}.$$

The reference values for p and α of
(5)$$\alpha_{0}=2.7,\;p_{0}=0.36$$
for carbon-group materials with the diamond lattice structure (Si and Ge) were determined in previous publications [68,91]. There, numerical studies were carried out and it was demonstrated that the breakup of Si/Ge nanowires into nanocluster ordered chains (α=2.4−3.0 and p=0.32−0.4) and the appearance of Si/Ge nanoclusters on Si/Ge nanowires were in very good agreement with the experimental data.

## 3. The Numerical Experiment Results and Their Analysis

Quantitative and even qualitative analysis of the physical mechanisms responsible for the development of the silicon-wafer surface morphology is rather complicated compared to the dynamics of an arbitrarily shaped nanocluster, which approaches the equilibrium state. In many cases, the equilibrium shape of a nanocluster is close to the so-called ‘*Wulff construction*’ corresponding to the minimum surface energy of the cluster for a given volume [103]. Naturally, the faces forming the Wulff construction have low surface energy densities. In silicon, the lowest surface energy density, *σ*, is achieved at face (111) [104]. The Wulff configuration for nanoclusters with a diamond cubic lattice is given by faces (111), (100), (311), and (110) and resembles a truncated octahedron:(6)$$\sigma_{\left(100\right)}/\sigma_{\left(111\right)}\approx 1.105,\;\sigma_{\left(311\right)}/\sigma_{\left(111\right)}\approx 1.125,\;\sigma_{\left(110\right)}/\sigma_{\left(111\right)}\approx 1.165.$$

Determination of the equilibrium shape of a nanocluster only from the condition of minimal surface energy does not take into account the actual kinetic processes on its surface (surface diffusion of bound atoms and re-deposition of free atoms from the near-surface region), which can significantly change the equilibrium shape of the nanocluster relative to the Wulff construction [68].

Under externally induced modification of the Si wafer surface, it is the kinetic processes both on its surface and in the near-surface region that determine the changes to its morphology. Depending on the characteristics of the ion flow hitting the target and the degree of its heating (temperature, if the surface is in a quasi-equilibrium state), one of the many “trajectories” of the system, along which the associated free Helmholtz energy decreases, is realized:(7)$$dF=dU_{surf}-TdS<0.$$

The wafer surface dynamics (unlike that for a single nanocluster where, as a rule, dS<0 for the decreasing total surface area and $dU_{surf}<0$) are accompanied by an increase in surface energy ($dU_{surf}>0,\;dS>0$ as the total surface area increases). Naturally, to satisfy inequality (7), minimal increases in surface energy are required during surface texturing. That is, the forming elements of the nanostructure should be largely confined by fragments of the “main faces” of the diamond cubic crystal lattice: (111), (100), (311), and (110). This is confirmed by the results of our numerical experiments. In discussing these, we focus on a qualitative analysis of the kinetic processes leading to the formation of nanostructures with different morphologies depending on the type of irradiated Si wafer faces.

Let us consider the stimulation of surface diffusion and re-deposition of atoms from the near-surface region of the free atom “vapor” (see Figure 1). The value of 2.7 for parameter α corresponds to a moderate wafer temperature [68]. The corresponding value of the parameter p, according to Equation (4), is 0.36. When the surface diffusion of atoms is induced by an ion beam, the parameter p increases to 0.54.

The substantial effect of this stimulation is seen when comparing the data of Figure 1 (configurations (a) and (b)) and Figure 2c obtained at the same moment in time. In our figures, atoms lying in the initial upper atomic layer with index j=0, atoms in the newly formed layers with index j>0, and atoms of the inner layers (j<0) are marked with different colors.

The role of the re-deposition effect is demonstrated in Figure 1 (configurations (c) and (d)) and Figure 3. Configurations (c) and (d) (Figure 1) are calculated at artificially reduced levels of atomic evaporation from the surface. If at some MC time step an individual atom has a chance to detach from the wafer surface, this detachment is realized with an additional probability filter $P_{filter}<1$ (the natural dynamics of nanostructuring corresponds to $P_{filter}=1$). Recall that the density of the reverse flux of atoms is not uniform because protrusions formed on the wafer intercept the depositing atoms by their tips, creating shadow zones around themselves. The shadow effect [105] increases with the height of these protrusions blocking the reverse flow of evaporated atoms to the surface of the craters formed. As a result, the reduction of the evaporating atom flux is accompanied by the self-consistent decrease in the protrusion heights and in the corresponding “pumping” of Si atoms from the craters of the formed nanostructure to its tips through the near-surface region (see Figure 1d).

Thus, the development of nanostructures on the surface of silicon wafers is associated with two types of mass transfer: the surface diffusion of bound atoms and the diffusion of free atoms in the near-surface region. The contribution of each process depends on the type of the crystallographic face that is exposed to external irradiation. The lower the surface energy density on a particular face (higher binding energy, ϵ, with nearest neighbors), the fewer silicon atoms evaporate from it. Therefore, surface diffusion is expected to dominate on the (111) face with a maximum value of ϵ ($\epsilon =\epsilon_{\left(111\right)}$), while diffusion in the near-surface layer of free atoms dominates during the structuring of the (110)-face with a minimum binding energy, $\epsilon_{\left(110\right)}$, of surface atoms. The results presented in Figure 1 confirm this assumption. The reduction in the surface diffusion in the configurations shown in Figure 1a,b and the evaporation of atoms in the configuration shown in Figure 1d (at a high rate of surface diffusion) dramatically suppress the development of nanostructuring compared to the data in Figure 2 and Figure 3.

Let us consider, in more detail, the formation of surface structural elements on the (111)-face (one such structure is shown in Figure 4). Each element is a trihedral truncated pyramid. There are three zones, whose roles in the growth of these truncated pyramids are significantly different.

The first zone is the set of relatively flat pit bottoms between neighboring pyramids. Single atoms ripped out of these bottoms are weakly bound to their nearest neighbors and chaotically move on the bottoms (such mobile “surface” atoms can be seen in configurations (e) and (f) of Figure 2). Finally, these atoms are captured by the foothills of the pyramids where the number of corresponding bonds is higher. The side faces of the formed pyramids comprise the second zone. Along those faces, the transport of trapped atoms is carried out from the first zone to the pyramid top. The third zone is a set of plateaus on the top of the formed pyramids. In these regions, new clusters are formed (see zone G in Figure 4).

Self-consistent processes in the indicated zones lead to the formation of nanostructures on the wafer surface only at sufficiently high levels of stimulated surface atom diffusion. With the corresponding increase in the parameter p, the relative change in the jump probability, P, in different zones of the surface equals (see Equation (2))
(8)$$dP/P=\left(m_{0}/p\right)\;dp.$$

The greatest intensification of surface diffusion occurs in zones with a large number of nearest neighbors, $m_{0}$, i.e., on the stepped slopes of the pyramids, and provides a sufficient influx of atoms onto the pyramid tops, so that new nuclei are formed (red arrows in Figure 4). This influx must exceed the downhill flow of weakly bound single atoms on the (111) face that find their way to the slopes from zone G (blue arrows in Figure 4).

Note that over time, small-scale pyramids (marked by vertical red arrows and blue ellipses in Figure 2c–f) are absorbed by larger ones. An analogous phenomenon (the well-known Ostwald ripening effect [106]) is observed in the dynamics of ridges/ ‘dunes’ during the structuring of the (110) and (112) faces (see Figure 3 and Figure 5).

Since almost flat plateaus on the top of the pyramids and bottoms of the pits (Figure 2) are well represented by (111)-type faces, the pyramid slopes form (100)-type faces with the second smallest surface energy density (see Equation (6)). The prediction of the structure of the ridge slopes formed on faces (110) is not so unambiguous (Figure 3). In the case of chemical etching, the formed ridges are oriented along the [$\bar{1}10$] axis, and their slope sides correspond to (111)-type faces [107]. In the case of the stimulated roughening transition that we have considered, the ridges are oriented along the [001] vector ([001]⊥[$\bar{1}10$]), and their slopes are represented by (100) and ($0\bar{1}0$) faces. The same structure of the ridges is also observed in experiments [108], in which the structuring of the surface (110) was induced by laser irradiation.

The reason for this ambiguity is related to the peculiarities of the kinetic processes determining the structuring of the Si wafer. Indeed, according to the thermodynamic reasoning, the structuring should be realized only with a minimum increase in the surface energy $U_{surf}$. Then, the slopes would have been (111)-type faces. The actual appearance of (100) faces seems unexpected because the surface energy density on such faces is higher than on (111) faces: $\sigma_{\left(100\right)}/\sigma_{\left(111\right)}\approx 1.105$ (see Equation (6)). This fact can be explained as follows. The angle between the (100) and ($0\bar{1}0$) faces/slopes is 90°, and the angle between the (111) and ($11\bar{1}$) facets is $70.5^{{}^{\circ}}$. For the same height of ridges, the total area of the faces of (111)-type, $S^{\left(111\right)}$, is higher than the total area of the faces of (100)-type, $S^{\left(100\right)}$. Specifically, $S{\;}^{\left(111\right)}/S^{\left(100\right)}\approx 1.34$ and
(9)$$U_{surf}^{\left(100\right)}=\sigma_{\left(100\right)}S^{\left(100\right)}<U_{surf}^{\left(111\right)}=\sigma_{\left(111\right)}S^{\left(111\right)}.$$

The dynamics of the ($\bar{1}\bar{1}2$) face (Figure 5) occurs in two stages. Initially, ridges are formed, one slope of which is a strip of (111) facet (see Figure 5a). On these strips, the already known instabilities presented in Figure 2 develop. At the later stages of its transformation, the shape of the modified surface resembles twisting sandy desert dunes [109,110], the vertices of which are fragments of the (111)-faces (see the upper inset in Figure 5). The gentle right-hand slopes (see the bottom inset of Figure 5) represent faces of the ($\bar{1}\bar{1}5$)-type with a large surface energy density. Forming just such slopes minimizes the increase in surface energy $U_{surf}$; the “loss” resulting from the increased average surface energy density on the ridge slopes is compensated for by the “gain” in their smaller total surface area. Similarly, the minimization of the surface energy increase was achieved by structuring the face (110) (see Figure 3 and Equation (9)).

The surface energy density, σ, is higher on face ($\bar{1}\bar{1}2$) than on face (110), $\sigma_{\left(\bar{1}\bar{1}2\right)}>\sigma_{\left(110\right)}$, and correspondingly, the binding energy is lower, $\epsilon_{\left(\bar{1}\bar{1}2\right)}<\epsilon_{\left(110\right)}$. Therefore, the evaporation/re-deposition effects play a more important role in the formation of nanostructures on this face (see Figure 5, right inset) than they do in the formation of such structures on face (110) (see Figure 1c,d and Figure 3). In particular, a feature appears in the dynamics of formed dunes. Specifically, when comparing configurations (b), (c), and (d) in Figure 5, their gradual drift from bottom to top is noticeable. This drift is related to the intense inhomogeneous exchange of the wafer surface with the near-surface region of vaporized atoms. On the one hand, the evaporation of atoms from craters between neighboring dunes is less intensive than from gentle ($\bar{1}\bar{1}5$) slopes. On the other hand, however, the return (re-deposition) of atoms to these craters is blocked to a great extent by shadow effects. Due to the resulting imbalance between the evaporation and re-deposition fluxes, atoms from steeper dune slopes drift in the near-surface layer to their gentler ($\bar{1}\bar{1}5$)-slopes, resulting in the observed dune drift depicted in Figure 5.

An important question arises of whether significant heating of the silicon wafer should be allowed when it is irradiated by ion flow. The theoretical answer is ambiguous. On the one hand, a positive factor is that at an elevated temperature, the processes of atoms’ evaporation and their surface diffusion are intensified. On the other hand, the distribution of the hopping probability of bound atoms along the possible hopping directions becomes more homogeneous (see Equation (3)). This homogeneity increases with *T* and suppresses the growth of nuclei of the new clusters formed at the tops of nanostructures, pyramids, ridges, and dunes, i.e., favors the detachment of the bound atoms from these nuclei. Note that this is the factor that restrains the effects of stimulated roughening transition in the nanostructuring of metal surfaces with an fcc crystal lattice [79].

The individuality of the structuring dynamics characteristics for silicon faces with different orientations is also reflected in the data presented in Figure 6. On face (111), where (as shown above) surface diffusion dominates, the dependence of $N^{\left(+\right)}\left(t\right)$ is close to linear (Figure 6a). For face (110) (with a lower binding energy of surface atoms), evaporation and further re-deposition of atoms are more prominent. Therefore, in the initial stage of surface modification (Figure 6b), the number of over-surface atoms, $N^{\left(+\right)}\left(t\right)$, increases sharply compared to the analogous dependence of Figure 6a. However, at later structuring stages, $t≳3\times 10^{6}$ MC steps, the value of $\partial N^{\left(+\right)}\left(t\right)/\partial t$ for face (111) is about twice as high as for face (110). The reason for the change of the leader in the creation of additional over-surface atomic layers is as follows. On the top plateaus of pyramids (see Figure 4), the atoms supplied easily create new clusters (fragments of (111) face), with the maximum atomic binding energy $\epsilon_{\left(111\right)}$. At the same time, the build-up of new layers on the rather sharp tops of ridges (see Figure 3) is readily seen to be difficult.

The data in Figure 6c clearly demonstrate two stages in the dynamics of (112)-face structuring, already discussed above. The slow accumulation of atoms in the over-surface layers ($t≲\hat{t}=2.5\times 10^{6}$ MC steps) culminates in the formation of single slopes/strips on (111)-faces (Figure 5a). The subsequent breakup of these strips into individual nucleation centers (see Figure 5, upper inset) substantially intensifies the growth of the dune-like nanostructure. Since at $t>\hat{t}$ multiple (111)-nucleation centers play a main role in the buildup of atomic over-surface layers, the dependence $N^{\left(+\right)}(t>\hat{t})$ shown in Figure 6c is quite similar in form to the dependence $N^{\left(+\right)}\left(t\right)$ for the (111) face texturing (see Figure 6a).

On the (100)-face, the surface nanostructure did not develop. In this case, tetrahedral pyramids with side (111)-faces should be formed. Moreover, at chemical etching [111], such pyramids are really observed. However, the laws of dynamics in the case of the stimulated roughening transition governing surface modification do not allow their appearance. The formation of tetrahedral pyramids is associated with a significant increase in the surface area of this nanostructure and a corresponding increase in the surface energy $U_{surf}$, which excludes the possibility of the system evolution along any “trajectory” with decreasing free Helmholtz energy.

Thus, we theoretically investigated various cases of silicon single-crystal surface structuring. In multi-crystalline silicon [112], both scenarios of surface modification analyzed above and a wide variety of their combinations are possible. Of course, in multi-crystalline samples, spectacular features found for ideal cases may be blurred or rendered implicit.

Hence, it would be useful to compare the emergence of nanostructures at single-crystal and multi-crystalline surfaces under the influence of comparatively low-entropy external factors stimulating various instabilities. This idea belongs to Erwin Schrödinger, who proposed that it is the Sun’s irradiation conducting low-entropy more-ordered energy to create biological structures and living organisms themselves [113,114]. He even put forward the interpretation that living beings are drawing negative entropy from the environment. Later on, Leon Brillouin equated negative entropy (‘negentropy’) to information [115]. Of course, this usage of terms is metaphorical, because physical entropy can only be positive or zero-valued [116]. Moreover, one can avoid the negentropy concept both in physics and in information theory without any loss of understanding [117,118]. We note that the initial solar irradiation is almost a blackbody one with a temperature equal to that of the Sun’s surface (approximately 5800 K). At the terrestrial surface, the effective temperature, $T_{eff}$, may differ from the initial blackbody temperature in certain directions and frequency intervals and is also influenced by the Earth’s atmosphere, oceans, and living organisms [119,120,121,122]. In any case, it may be asserted that $T_{eff}$ is much higher than the temperature of the Earth’s infrared radiation into the open space. Thus, the steady-state situation is highly non-equilibrium and an outflow of entropy from the Earth exists. The overall trend in the behavior of our planet consists of the maximum possible entropy production [123,124,125], which differs from the previously suggested minimum entropy production in the non-equilibrium stationary state [116]. The latter principle is at least restricted to a region close to equilibrium, whereas the former one is universal and describes the highly non-linear regime [126]. The maximum value is achieved for inherent system constraints.

The indicated thermodynamic considerations are quite general. When applied to the polychromatic solar light, they do not reveal a direct way to induce periodic patterns in inanimate nature and biota by the direct action of irradiation. Observations confirm this conclusion. Nevertheless, the conversion of incident solar rays into other convenient forms of energy is ubiquitous in plants and photovoltaic devices [127] and is more or less understood from the thermodynamic viewpoint [128].

The situation is quite different for the irradiation of metal, semiconductor, and insulator surfaces by artificial, almost monochromatic laser light. In this case, periodic patterns emerge, having in general slightly smaller spatial structure periods than the laser wavelength λ [17,129,130]. For some metallic targets, the hierarchical state with two sets of periodical structures are observed, with the additional period being about λ/10 [131]. It is important, however, that the smaller component of the double-period patterns was not found for silicon [131]. On the other hand, for certain laser irradiation regimes, silicon surfaces reveal simultaneous formation of structures with similar periods but different appearances: ripples and grooves [132,133]. Laser-induced structures with a period significantly smaller than λ were also obtained on tungsten surfaces [134]. An unequivocal theoretical explanation of those phenomena is still lacking [135,136], probably because the elementary kinetic processes were not previously studied. As we showed above, such studies are necessary to understand the formation of nanostructures as the basis of microstructures in the induced complex hierarchical textures.

We are interested in applying another external factor disturbing solid surfaces from the equilibrium condition. Specifically, in our experimental studies described below, the ion flow in the high-density trampoline regime [20,90] is shown to be such a factor, which successfully modifies the impacted surfaces in a gentle way, in accordance with our calculations presented above.

## 4. Experimental Texturing of Silicon Wafers for Photovoltaics

Conventional chemical technologies producing black silicon structures for photovoltaics are environmentally unfavorable [137,138]. This is especially true for the metal-catalyzed chemical etching (MCCE) technology intended to create these structures on the surface of multi-crystalline silicon wafers, which comprise the most promising material for future solar cells [112,139,140]. Problems of environmental safety can be solved by using ion-plasma technologies to modify the surfaces of silicon wafers to create black silicon structures. The most efficient in terms of performance and the ability to control spectral characteristics of the light reflection coefficient, R, is the trampoline technology of collective ion-plasma sputtering of solids [20,90]. We note that along with reducing the load on the ecosystem, the trampoline technology, having a high productivity, is compatible with technologies for cutting silicon ingots into wafers. Therefore, it is possible to combine the cutting of ingots into wafers, the removal of a defective layer after cutting, and the creation of black silicon structures both on single-crystal c-Si and multi-crystalline mc-Si wafers in one production cycle.

Experiments on modifying the silicon wafer surface to the black silicon state were carried out using the helicon source, a schematic representation of which is shown in Figure 7. The helicon discharge was excited using radio frequency, RF, power delivered to the flat antenna through a matching device NAVIO. The RF field was created using the CESAR RF generator at a frequency of 13.56 MHz and a power of 1200 W. The antenna was located on a quartz window. The magnetic system provided the helicon plasma generation near the window and the creation of an accelerated ion-plasma flow in the direction of the substrate holder. The vacuum chamber pumping was carried out using the pump STP-A803C of Edwards firm. The oil-free spiral pump XDC35i of the same firm provided the preliminary evacuation. The residual chamber pressure was measured using the ionization vacuum gauge VIT-2. The pressure was measured by vacuum detector CMX OM 25 Brooks Instrument with the measurement range of 33 Pa. The substrate holder contained a built-in heater with a thermocouple temperature meter. Argon was the plasma-forming gas; its pressure was maintained in the range of 1 Pa. A detailed diagram of a technological chamber with a helicon source and a description of its operation modes can be found in [90]. The parameters of plasma and ion-plasma flow were determined using a Langmuir probe and a five-electrode electrostatic analyzer. The structure of the surface after its ion treatment was studied using field emission scanning electron spectroscopy. Reflection coefficients were measured using the Loana system of PV-tools firm.

The trampoline mode is realized when the threshold values (varying for different impacted materials [90]) of the ion density in the ion-plasma flow onto the target $j_{+}\approx 10\;\mathrm{mA}/{\mathrm{cm}}^{2}$ and the ion energy $E_{i}=50\;\mathrm{eV}$ are exceeded. This mode is characterized, in particular, by the peculiar hierarchical morphology of the sputtered surfaces, including nano- and micro-structures [20]. With regard to silicon, the effective creation of optically black silicon structures, which are grossly homogeneous over the wafer surface, is realized if the latter is pretreated in such a way that seed nanoscale surface structures exist on it from the outset. This is realized either via short-term exposure to an ion-plasma flow in the trampoline mode or by plasma-chemical etching in an Ar + SF_6_ gas mixture [141,142]. In this respect, the formation of starting nanoscale structures is more critical for c-Si wafers with a more uniform surface than for their mc-Si counterparts.

As is shown in Figure 8, an increase in the fluence of the ion-plasma flow with parameters corresponding to the trampoline mode (panels a and b) first leads to the texturing of regions with low inter-crystalline bond energies in the zone of surface defects formed during crystal polishing. At even higher fluences (panel c), structures develop on the initial seed inhomogeneities. This behavior may be caused by additional re-deposition of sputtered ionized silicon atoms under conditions when the excited silicon atoms return to the ground state with the emittance of characteristic radiation. An increase in the ion energies $E_{i}$ in the flow or heating of the wafer leads to significant growth in the size of surface structures up to microns and to the acceleration of the texturing processes. At the same time, nanoscale structures are preserved either as modulations of micron and submicron formations or in the form of built-in structures in larger structures. Such two-scale hierarchical structures are not only inherent to the textured silicon but are also formed on metallic surfaces [20]. Our numerical simulations of the processes stimulated by ion irradiation (see Section 3) demonstrate that, in the initial stage, self-organization of nanosized structures occurs. These nano-structures further develop during the subsequent trampoline texturing, i.e., under high-density low-energy ion irradiation. Surface structures emerge regardless of the spatial orientation of the crystallographic axes in the near-surface silicon wafer layers. Thus, the trampoline mechanism for complex texturing of the surface of mc-Si wafers shows its effectiveness.

In Figure 9 (panel a), the initial surface structure of the mc-Si wafer is displayed. Figure 9b shows the surface ordering of this wafer due to plasma-chemical etching. Figure 9c demonstrates the subsequent creation of the desired black silicon structure after the action of an ion-plasma flow in the trampoline mode. The same result as in Figure 9c was achieved when the small-fluence ion-plasma processing provided the initial ordering of the surface. In both cases, the processed surface of the mc-Si wafer, as is shown in Figure 10, demonstrates a high absorption efficiency of solar energy across a wide range of visible light wavelengths. Therefore, one can avoid chemical etching while fabricating black silicon surfaces of good quality, which, as was emphasized above, is environmentally friendly. The effective structuring of mc-Si wafers to the state of black silicon in the trampoline regime of the ion-flow irradiation is consistent with the results of numerical experiments presented in Figure 2, Figure 3 and Figure 5.

As follows from the data presented in Figure 11, during the trampoline texturing of c-Si wafer surfaces, hierarchical nano- and micro-structures are created, maintaining the light reflection coefficient down to 2%. While carrying out experiments with black silicon production with the use of the trampoline mode of the ion-plasma flow, we noticed several features. Specifically, (i) with an increase in $j_{+}$, the dependence of the sample’s reflection coefficient, R, on the light wavelength becomes uniform demonstrating a tendency to its decrease in the region of long waves; (ii) R weakly depends on $j_{+}$ for short wavelengths; (iii) R increases with $E_{i};$ (iv) preheating of the substrate leads to a decrease in R and to its uniform dependence on the wavelength; and (v) R increases with the processing duration.

All those features make it possible, using the trampoline technology, to effectively control the spectral dependence of R over a wide range of wavelengths and create photocells with optimal parameters for operation in the Earth’s atmosphere, as well as in the near and far space. At the same time, to explain the details of physical processes leading to the peculiarities concerned, we are going to carry out further numerical simulations based on the already understood concepts.

## 5. Discussion of the Results

In this section, we correlate our theoretical model, general ideas, and results of numerical calculations with the experimental data on black silicon. We also discuss this range of issues in a more global sense.

First, it is important to emphasize that preheating makes surface atoms more mobile, and in particular, some of these atoms vaporize so that the density of defects increases. This circumstance provokes the appearance of heterophase fluctuations and roughening transition in the surface layers of the target under the influence of dense ion flows. After irradiation, the hierarchical texture in silicon is frozen (making it optically black) and looks somewhat similar to the theoretically calculated patterns (see Section 3). The difference may be due to the simplicity of the model. In particular, the latter does not take into account the ionization of the silicon surface atoms by impinging Ar ions. This consideration is confirmed by a much larger similarity of our theoretical periodic structures and those induced on the silicon surface by laser light [48,130,131,132,133] when local heating is the main factor leading to surface instability. Depending on the laser irradiation parameters, ripples, grooves, and/or spikes can grow on the target surfaces. Nevertheless, the ion-flow pulses in the trampoline regime [20,90,142] ensure the richer several-scale picture shown in Figure 11a. Although patterns are readily seen using an electron microscope, in the visible light range, the inhomogeneities are averaged out and the wafer surface becomes optically uniform with very small reflectivity in the whole wavelength range.

We emphasize that the conventional pre-melting [50,143], e.g., the existence of the quasi-liquid layer on the water ice surface below the bulk melting point [144,145,146,147,148], confirming a hypothesis of Michael Faraday [149], occurs in the equilibrium situation, whereas targets impacted by ions or laser beams are in the non-equilibrium state. At the same time, both kinds of perturbed surfaces are patchy [145,147,150,151] since they are patterned by heterophase fluctuations [51,52]. Molecular dynamic studies [152] revealed that in actual truth, the Faraday quasi-liquid layer consists of several partially melted ice layers. Thus, those layers are unstable against roughening transitions (see [143] and Section 3). Since surface melting, premelting, and heterophase fluctuations are general phenomena, being not restricted to water (although the latter is rich in surprise [148]), they may emerge on surfaces of other materials. Indeed, the phenomena concerned were observed in Cu [153], Pb [154], and Ni [155]. Our experiments indicate that similar processes occur on silicon surfaces as well but in the non-equilibrium set-up. The non-equilibrium quasi-liquid states are created under the influence of dense ion-plasma flows and then frozen as peculiar hierarchical structures that cannot be obtained otherwise.

Heterophase fluctuations of the surface-type described above were also suggested to exist in the solid bulk, where many-body orderings occur below some temperatures, e.g., superconductivity or charge density waves. There, patches (puddles) of the enhanced or reduced order parameter were assumed, in agreement with the experiment for certain superconductors [156,157]. This model is a moderate version of that one, anticipating a nonhomogeneous percolating order parameter network in superconductors with small coherence lengths [158,159].

We observed a predominantly two-scale hierarchical surface structure of black silicon wafers and confirmed the emergence of such structures via calculations. Analogous structures were obtained for a large number of artificially processed materials [11,12] and exist in animals and plants [160]. In our case, those combined, well-developed nano- and micro-structures are produced due to the high density of the ion flow in the trampoline mode. A texture of this kind on the surfaces of silicon wafers made it possible to obtain the sought optical properties (see panel b of Figure 10) suitable, in particular, for photovoltaic applications.

## 6. Conclusions

Our treatment of silicon surfaces using ion flows in the trampoline mode made it possible to create black silicon surfaces in an industrially and environmentally friendly manner. The reflection coefficients in the range 300–800 nm are 4–5% for mc-Si and 2–4% for c-Si wafers (see Figure 10 and Figure 11, respectively). This is slightly worse than the results of 1.5–2% obtained using the MCCE technology [161,162]. However, our results are industrially competitive [1,2]. On the other hand, an important advantage is the environmental friendliness of the involved processes, especially in comparison with the MCCE method. The absence of environmental pollution makes it possible to combine in one production process the crystal growth, cutting of ingots, removal of the defective layer, and the creation of a black silicon structure, which results in a new market product with benefits—black silicon wafers without a defective layer. The performance of the proposed method is consistent with that of existing ingot-cutting equipment, and the cost is lower. From the viewpoint of fundamental physics, our results are no less important because a method of creating a new non-equilibrium quasi-liquid state was presented and the microscopic kinetics of the corresponding processes were theoretically elucidated.

## Figures and Tables

**Figure 1 nanomaterials-13-02715-f001:**
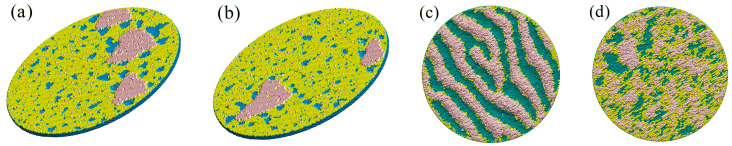
Influence of stimulated surface atom diffusion and reattachment effects of evaporated atoms on the surface morphology dynamics of silicon wafers. The initial surface layer of atoms is marked in yellow, and the layers above, in metallic pastel. Configurations (**a**,**b**) show the (111)-facet structure in the absence of surface process stimulation in two random realizations: α=2.7, p=0.36, disk diameter D=180 (≈70 nm), and $t=3\times 10^{6}$ MC steps (compare this structure to the one shown in Figure 2c). Configurations (**c**,**d**) demonstrate the (110)-facet structure at different levels of atom sublimation blockage: α=2.7, p=0.54, and D=180. For configuration (**c**), $P_{filter}=0.2$, $t=3\times 10^{6}$ MC steps, number of atoms above the initial surface is ~1.6 times smaller than at $P_{filter}=1.0$—see Figure 3. For configuration (**d**), $P_{filter}=0.05$ and $t=6\times 10^{6}$ MC steps. $P_{filter}$ is the additional probability of an atom to be detached (see further details in the main text).

**Figure 2 nanomaterials-13-02715-f002:**
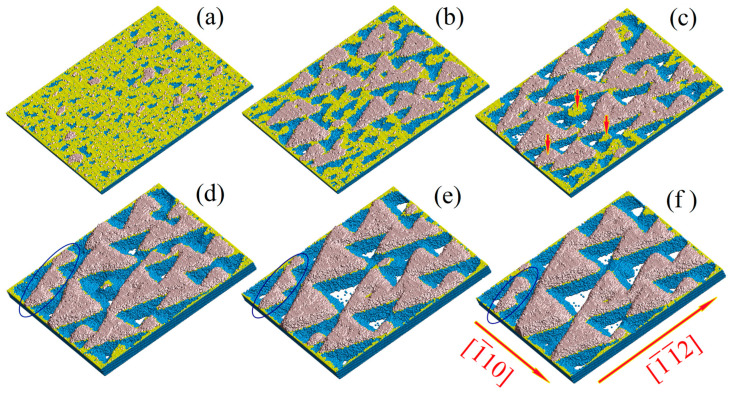
Surface dynamics on the $\left(111\right)$-face of silicon under stimulation of surface processes: α=2.7 and p=0.54. The computational domain size is 190×170 (~73 nm×65 nm) along $\left[\bar{1}10\right]$ and $\left[\bar{1}\bar{1}2\right]$ axes, respectively. Configurations (**a**–**f**) correspond to simulation times $t=3\times 10^{5}$, $1.5\times 10^{6}$, $3\times 10^{6}$, $5\times 10^{6}$, $7.5\times 10^{6}$, and $9\times 10^{6}$ MC steps, respectively. Red vertical arrows (configuration (**c**)) and blue ellipses (configurations (**d**–**f**)) indicate clusters absorbed by their neighbors with time. Atomic layers forming pit bottoms are highlighted in white to clearly represent the developed nanostructure morphology.

**Figure 3 nanomaterials-13-02715-f003:**
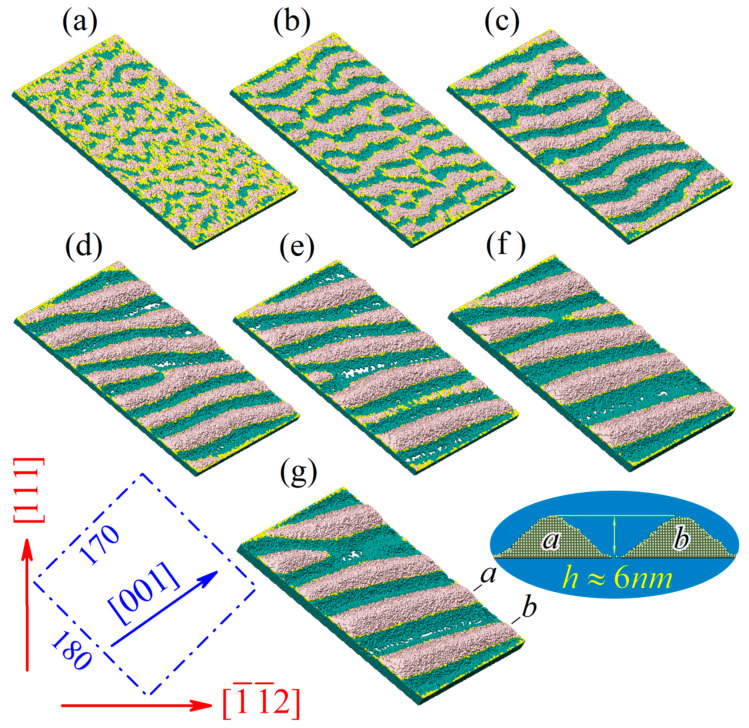
Surface dynamics on the $\left(110\right)$-face of silicon under stimulation of surface processes: α=2.7, p=0.54. Configurations (**a**–**g**) correspond to simulation times $t=10^{5}$, $3\times 10^{5}$, $10^{6}$, $3\times 10^{6}$, $6\times 10^{6}$, $9\times 10^{6}$, and $12.5\times 10^{6}$ MC steps, respectively. The bottom-left inset shows the orientation in space and the size of the wafer surface. The bottom-right inset represents the cross-section of two neighboring ridges. The full process is demonstrated in the “110-Ridges.mp4” video file—see the Appendix A.

**Figure 4 nanomaterials-13-02715-f004:**
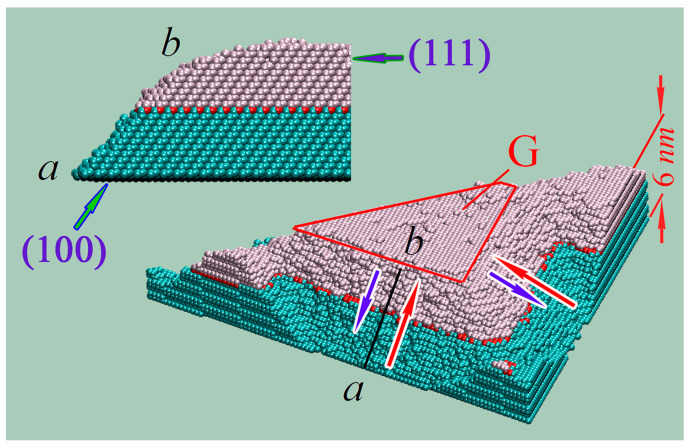
Characteristic structural element formed on the $\left(111\right)$-face by the time $t=9\times 10^{6}$ MC steps (see Figure 2f). Initial surface layer atoms (layer index j=0) are marked in red, atoms of the layers below (j<0) in dark cyan, and atoms of the newly developed layers in metallic pastel. Blue and red arrows point in the direction of surface diffusion flows of the atoms: down- and uphill on the structural element slopes, accordingly. The new layer formation zone, G, is marked by red line segments. The top-left inset shows the cross-section of the structural element slope along the a−b line segment.

**Figure 5 nanomaterials-13-02715-f005:**
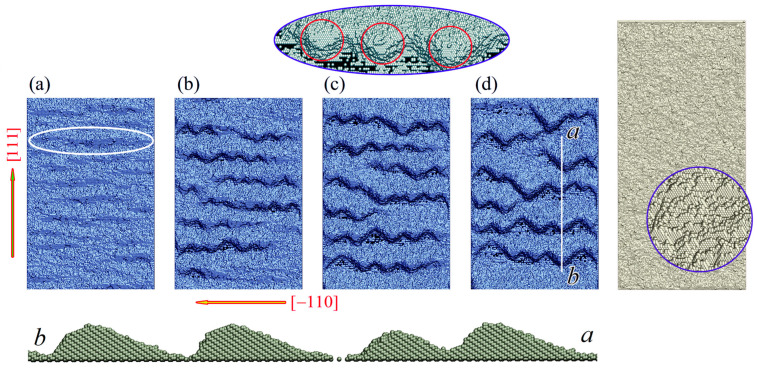
Morphology evolution of the $\left(\bar{1}\bar{1}2\right)$-face of silicon under stimulation of surface processes: α=2.7 and p=0.54. The size of the presented domain is 160×240 (~61 nm×92 nm) along $\left[\bar{1}10\right]$ and [111] axes, respectively. Configurations (**a**–**d**) correspond to simulation times $t=2.5\times 10^{6}$, $4\times 10^{6}$, $6\times 10^{6}$, and $8\times 10^{6}$ MC steps, respectively. The ellipse in configuration (**a**) highlights one of the formed strips of (111)-type. The top inset shows the leading tips of the developing dunes (fragments of the (111)-face inclined to the plane of the wafer). The bottom inset represents the cross-section of near-surface layers along the a−b line segment marked in configuration (**d**). The right inset demonstrates the surface morphology of the wafer (160×340, ~61 nm×131 nm) at the time $t=8\times 10^{6}$ MC steps, for which the sublimation was effectively blocked ($P_{filter}=0.05$). The structure of the surface, a magnified fragment of which is highlighted by a blue circle, resembles fish scales chaotically formed from (111)-face elements. The full process is demonstrated in the “112-Dunes.mp4” video file—see the Appendix A.

**Figure 6 nanomaterials-13-02715-f006:**
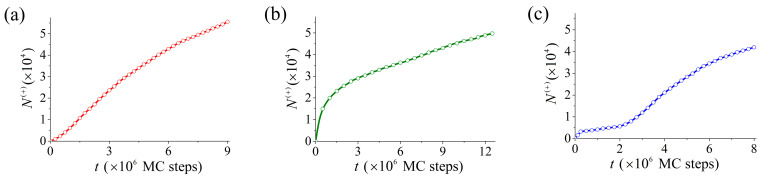
Change in the number of atoms, $N^{\left(+\right)}\left(t\right)$, over time in nanoclusters, formed above the initial faces. Dependencies presented in graphs (**a**–**c**) correspond to the surface nanostructuring processes shown in Figure 2, Figure 3 and Figure 5. Values $N^{\left(+\right)}$ are normalized for an area of 100×100 ($38.4\times 38.4\;{\mathrm{nm}}^{2}$).

**Figure 7 nanomaterials-13-02715-f007:**
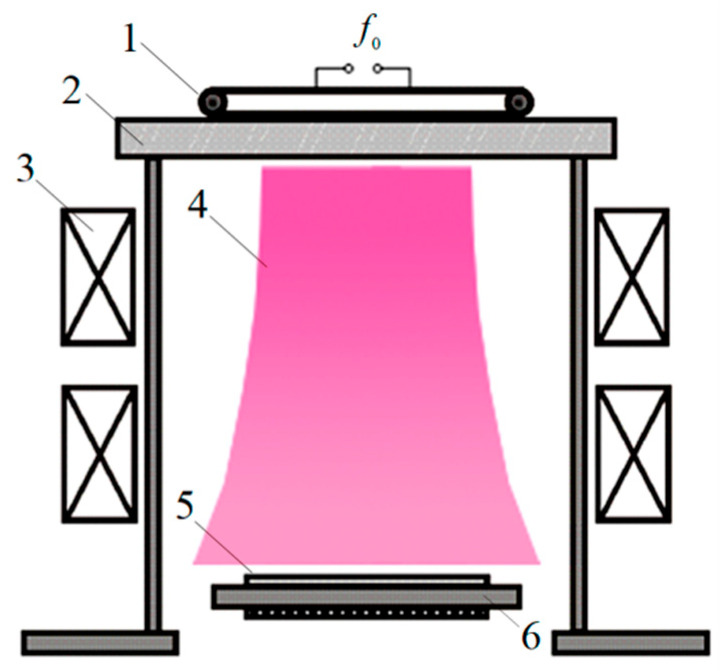
A principled scheme of the ion-plasma helicon source of the ion plasma flow: (1)—antenna; (2)—dielectric window; (3)—magnetic system of the helicon source; (4)—helicon source compartment; (5)—substrate; and (6)—substrate holder.

**Figure 8 nanomaterials-13-02715-f008:**
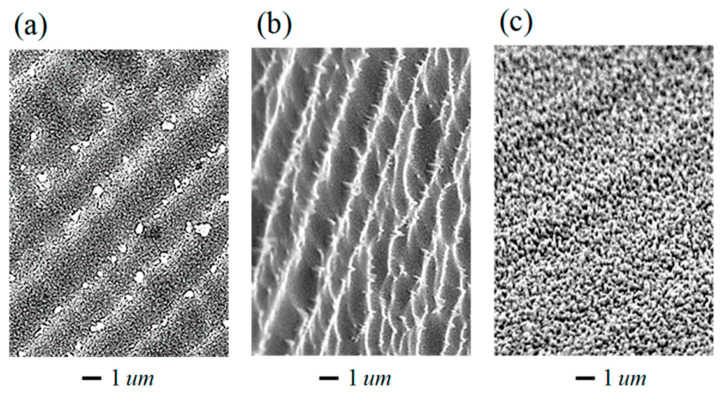
Scanning electron microscopy, SEM, micrographs of single-crystal silicon structures for fluences of the ion-plasma flow, corresponding to different duration of treatment in minutes: (**a**)—0.5; (**b**)—1.0; and (**c**)—5.0.

**Figure 9 nanomaterials-13-02715-f009:**
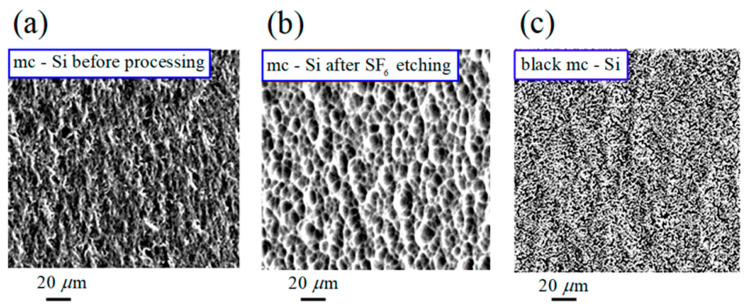
SEM micrographs of mc-Si wafers for different types of processing: (**a**) the initial surface structure; (**b**) the surface structure ordering because of SF6 plasma-chemical etching; and (**c**) the black silicon surface structure after the action of an ion-plasma flow in the trampoline mode.

**Figure 10 nanomaterials-13-02715-f010:**
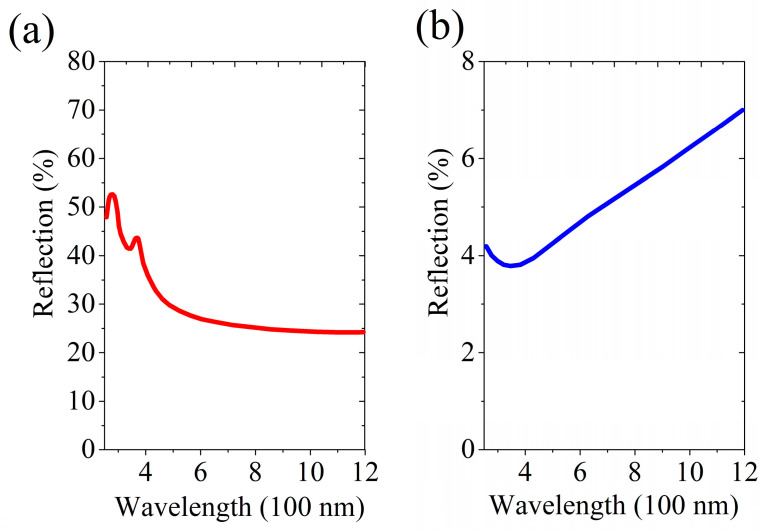
Dependence of the reflectance, R, for the mc-Si wafer on the light wavelength in the visible range: (**a**) back side of the wafer, and (**b**) processed side of the wafer with the black silicon structure.

**Figure 11 nanomaterials-13-02715-f011:**
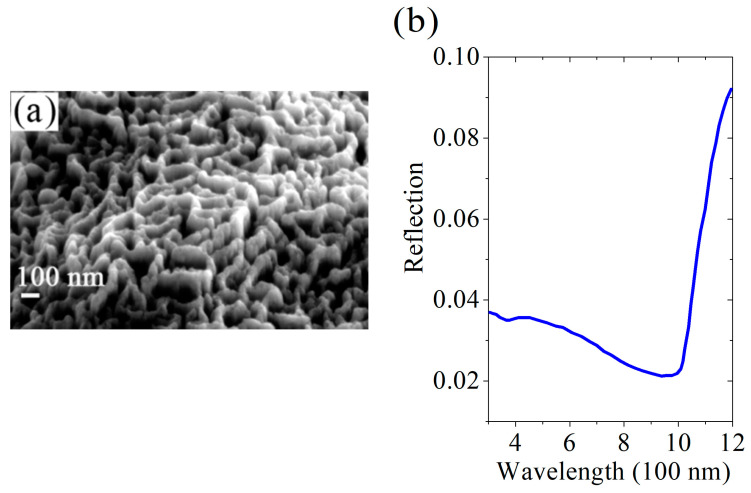
C-Si wafer: (**a**) SEM micrograph of the surface hierarchical structure on the processed black silicon wafer; (**b**) spectral dependence of the reflection coefficient R of this wafer.

## Data Availability

Data will be made available on request.

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
