# Peer review of "Hierarchical Structuring of Black Silicon Wafers by Ion-Flow-Stimulated Roughening Transition: Fundamentals and Applications for Photovoltaics"

_nanomaterials, 2023, doi:10.3390/nano13192715_

Round 1

Reviewer 1 Report

Some suggestions or issues to the manuscript.

1. The summary of the "Introduction" section is not concise enough and the description is cumbersome.

2. There are too many references cited, it is recommended to only cite those directly related to the research content of the paper.

3. In the section "Experimental texturing of silicon wafers for photovoltaics", detailed experimental information needs to be introduced clearly.

4. “Discussion of the results and conclusions” are suggested to be divided into two sections.

5. Experimental results require more characterization methods or data.

Reviewer 2 Report

The authors review the works on forming “black” silicon by surface re-arrangement through ion-flow numerically and experimentally. This review is comprehensive and complete and is worthy of publication. However, I have some suggestions and questions before recommending it to be accepted for publication. 

1.     The word “black” is an adjective, and it is hard to define the level of reflectance that is black enough. 20 % reflectance could be black enough for some people. Therefore, I suggest it is better not to use the word “black” silicon throughout the manuscript.

2.     As far as I know, the MACE method could achieve much lower reflectance than the ion flow. Is it possible to compare the pros and cons of the two methods? 

3.     Can the numerical simulation simulate the reflectance? And what is the correspondence of the numerical structure to the experimental results? Do they have a similar structure? If it is, please identify it. If it is not, why?

4.     The surface morphology from SEM shown in Fig. 11 achieves 0.02 % reflectance at around 1000nm wavelength, which is 300 times reduction as compared to the reflectance shown in Fig. 10. Is it a migrative reduction or a dramatical reduction? 

Round 2

Reviewer 1 Report

The manuscript can be accepted

Reviewer 2 Report

The authors addressed all my comments. I recommend this manuscript for publication.